# Association between SARS-CoV-2 Viral Load and Patient Symptoms and Clinical Outcomes Using Droplet Digital PCR

**DOI:** 10.3390/v15020446

**Published:** 2023-02-05

**Authors:** Elizabeth Hastie, Harold Amogan, David Looney, Sanjay R. Mehta

**Affiliations:** 1Division of Infectious Diseases and Global Public Health, University of California San Diego, La Jolla, CA 92039, USA; 2Veterans Medical Research Foundation, San Diego, CA 92161, USA; 3San Diego Veterans Affairs Medical Center, Department of Medicine, San Diego, CA 92161, USA

**Keywords:** COVID-19, SARS-CoV-2 viral load, droplet digital PCR, RNA quantification

## Abstract

The association between nasopharyngeal (NP) SARS-CoV-2 viral loads and clinical outcomes remains debated. Here, we examined the factors that might predict the NP viral load and the role of the viral load as a predictor of clinical outcomes. A convenience sample of 955 positive remnant NP swab eluent samples collected during routine care between 18 November 2020 and 26 September 2021 was cataloged and a chart review was performed. For non-duplicate samples with available demographic and clinical data (i.e., non-employees), an aliquot of eluent was sent for a droplet digital PCR quantification of the SARS-CoV-2 viral load. Univariate and multivariate analyses were performed to identify the clinical predictors of NP viral loads and the predictors of COVID-19-related clinical outcomes. Samples and data from 698 individuals were included in the final analysis. The sample cohort had a mean age of 50 years (range: 19–91); 86.6% were male and 76.3% were unvaccinated. The NP viral load was higher in people with respiratory symptoms (*p* = 0.0004) and fevers (*p* = 0.0006). In the predictive models for the clinical outcomes, the NP viral load approached a significance as a predictor for in-hospital mortality. In conclusion, the NP viral load did not appear to be a strong predictor of moderate-to-severe disease in the pre-Delta and Delta phases of the pandemic, but was predictive of symptomatic diseases and approached a significance for in-hospital mortality, providing support to the thesis that early viral control prevents the progression of disease.

## 1. Introduction

The correlation between severe acute respiratory syndrome coronavirus 2 (SARS-CoV-2) viral loads and coronavirus disease 2019 (COVID-19) outcomes has been heavily investigated since the beginning of the pandemic. Determining the utility of an accurately determined SARS-CoV-2 viral load in optimizing early treatment eligibility, preventing transmission by identifying asymptomatic carriers and infectiousness after the disease, and determining the prognosis remain open questions [1,2,3]. Studies have evaluated the correlations between the viral load and the transmissibility, vaccination status, age, comorbidities, and disease severity [4,5,6,7,8,9,10,11,12,13,14,15,16,17,18,19,20,21,22,23,24]. These studies, however, have generated conflicting results. Although a few studies have revealed a correlation between the viral load and the vaccination status, disease severity, and transmissibility [5,8,16,18,22,23,25,26], others have shown no statistical significance between the viral load and the disease severity [4,17,19,20] or the vaccination status [7,9,10,21]. Even a recent (2022) systematic review of 34 such studies evaluating the relationships between the disease severity and the viral load found that a similar number of studies supported and opposed this hypothesis [12]. The viral load has also been useful in determining the efficacy of novel treatments. The Blaze-1 trial, for example, utilized the viral load as its primary end point and determined that the combination therapy bamlanivimab and etesevimab resulted in a statistically significant reduction in the SARS-CoV-2 viral load at day 11 [15].

The vast majority of studies looking at the predictive impact of the viral load, including the Blaze-1 trial, utilized quantitative reverse transcription polymerase chain reaction (RT-qPCR) assays to detect SARS-CoV-2 RNA and generate a viral load [4,6,7,9,10,12,14,15,16,17,18,20,21,22,23,26]. In RT-qPCR, quantification is achieved by comparing the cycle threshold (Ct) values or the PCR cycle at which the fluorescence intensity reaches a specified threshold with standard curves created from samples with known quantities [27]. This generated result is dependent on standard calibration curves and is not an absolute quantification [3,27,28,29]. Although many assays use RT-qPCR for the diagnosis of COVID-19 [1,2,29,30,31,32,33], a droplet digital polymerase chain reaction (ddPCR) is newer quantitative PCR technology. Available since 2011, ddPCR allows for the absolute quantification of nucleic acid [27,34,35] without the need for a standard curve [29,30,32,33,35]. Multiple studies have compared ddPCR with RT-qPCR and have consistently found ddPCR to be more sensitive, specific, and repeatable [1,2,3,28,29,30,31,32,35,36,37,38]. This is particularly true in samples with a low viral load, in which RT-qPCR has been shown to produce false-negative results [2,29,35,39]. Although qPCR is currently more ubiquitously available, with these advantages, a few anticipate that ddPCR will overtake RT-qPCR as the new gold standard in future pandemics [32]. The disadvantages of ddPCR include an increased cost (estimated to be about 5–10% more) and an increased analytic time (estimated to be about 2 h longer) [29,30,32,39]. If the sample is not properly diluted, ddPCR may also underestimate the viral load when it is very high as the individual droplets of the PCR reactions are oversaturated [29].

In this study, we sought to explore the associations between SARS-CoV-2 viral loads and patient symptoms, demographics, and clinical outcomes of COVID-19 utilizing ddPCR. With prior studies utilizing RT-qPCR revealing conflicting results, we sought to explore the associations utilizing the more sensitive, specific, and reliable ddPCR. A more reliable association between the viral load and disease could aid earlier risk stratification and prognostication as well as the optimization of treatment eligibility and limiting transmissibility.

## 2. Materials and Methods

### 2.1. Patients and Specimens

A convenience sample was obtained consisting of 955 remnant nasopharyngeal (NP) swab eluent samples from positive tests collected during routine clinical care at the San Diego Veterans Affairs Medical Center between 11/18/20 and 9/26/21. All testing was obtained from the emergency room or from ancillary outdoor testing facilities set up outside the hospital and regional clinics.

### 2.2. Patient Consent Statement

The current study was approved by the Research Ethics Committee of San Diego Veterans Affairs Medical Center. The requirement to obtain informed consent was waived by the Ethics Committee.

### 2.3. SARS-CoV-2 RNA Testing

NP swabs were collected by trained nurses at testing locations and placed in 3 mL of a universal transport medium. Initial testing for the presence of SARS-CoV-2 was performed using several different platforms, including Cepheid GeneXpert^®^ (Sunnyvale, CA, USA), Roche Liat^®^ (Basel, Switzerland), Roche COBAS 6800^®^ (Basel, Switzerland), and BioFire^®^ (Salt Lake City, UT, USA). The remnant samples from the positive tests were aliquoted and stored at −80 °C until a further analysis was performed.

### 2.4. Droplet Digital PCR

RNA was extracted from the samples using a QIAamp^®^ Viral RNA Mini kit (Qiagen, Hilden, Germany). Following the kit’s protocol, RNA was extracted from 140 µL of the sample and eluted from the QIAgen column with 60 µL of buffer AVE. The nucleic acid concentration was then measured using a NanoDrop^®^ (Thermo Fisher Scientific, Waltham, MA, USA) instrument.

Bio-Rad’s One-Step RT-ddPCR Advanced Kit^®^ (Hercules, CA, USA) for the probe protocol was used to estimate the viral load. For ddPCR, each sample was prepared in duplicate with the average reported as the viral load. Briefly, 1 µL of the sample was loaded into a reaction mix. Droplets were generated from the reaction mix using a Bio-Rad QX200 Droplet Generator and PCR was run overnight using Bio-Rad’s C1000 Touch thermal cycler. The PCR results were then read on a Bio-Rad QX200 Droplet Reader the following day.

The oligonucleotide sequences, PCR settings, and thermal cycler conditions are shown below in Table 1, Table 2 and Table 3.

### 2.5. Clinical Data Collection

A chart review was conducted by EH and SRM to extract the demographic and clinical details for each included sample. Duplicate samples (e.g., repeat testing from the same individual) were excluded, with only the first positive test kept for the analysis. The samples from the employees of the Veterans Affairs (VA) Medical System were also excluded as full demographic and clinical data were not available for these individuals. Occasionally, subsequent care was obtained outside the VA medical system, but summarized records of these visits were usually available in the VA electronic medical record.

### 2.6. Statistical Analysis

The statistical analysis was performed using the R statistical computing program. Associations between the continuous variables were examined using parametric (Pearson correlation) and non-parametric (Spearman correlation) approaches. The differences between the medians across the groups were compared in a pairwise fashion using the non-parametric Mann–Whitney test, given that the SARS-CoV-2 RNA loads were non-normally distributed. Two-sided exact *p*-values were reported; *p* < 0.05 was considered to be statistically significant. A multivariate linear regression was performed using the glm function in R. The Akaike information criterion (AIC) was used to compare the multivariate models and determine which one was the best fit for the data (i.e., explaining the greatest amount of variation with the fewest number of variables).

## 3. Results

After removing the duplicate samples and the samples collected from employees, a total of 698 veterans were included, 86.6% (*n* = 605) of whom were male. The average age of the individuals included in the study was 50 years (range: 19–98). Sampling occurred at a median of 4 days after the onset of symptoms (range: −7 to 84). Three-quarters of individuals (76.3%, *n* = 529) were unvaccinated at diagnosis and 21.1% (*n* = 146) were fully vaccinated at the time of diagnosis with a SARS-CoV-2 infection. The majority of individuals were vaccinated with the Pfizer vaccine (58%, *n* = 95), followed by the Moderna vaccine (31%, *n* = 50) and the Johnson and Johnson vaccine (11%, *n* = 18). The frequency of comorbid conditions are shown in Table 4.

The majority of individuals developed symptoms, with 75.3% (*n* = 516) reporting respiratory symptoms, 46.4% (*n* = 316) a fever, 33.8% (*n* = 231) GI symptoms, and 23.1% (*n* = 161) a loss of taste and/or smell. A total of 28.9% (*n* = 198) individuals had both respiratory and GI symptoms. A total of 19.1% veterans were evaluated only in the emergency department, 21.3% (*n* = 149) were admitted to the hospital, and 8.6% (*n* = 60) required ICU-level care; 33.3% of these (*n* = 20) required intubation and 16 individuals died during hospitalization.

### 3.1. Associations with SARS-CoV-2 Nasopharyngeal Viral Load

We first evaluated the association between the SARS-CoV-2 log10 viral load and the timing of sampling in relation to the onset of symptoms; a strong correlation was observed (Spearman rho = −0.39; *p* < 0.0001) (Figure 1). We also compared the log_10_ viral loads in the samples before the Delta wave (sampled prior to 1 July 2021) with those during the Delta wave (sampled on 1 July 2021 or later) and we found that the viral loads were higher during the Delta wave (3.65 vs. 5.03; *p* < 0.0001). We then examined the association that the SARS-CoV-2 NP viral load had with the known risk factors for severe COVID-19. In these univariate analyses, the presence of hypertension and diabetes mellitus was not associated with differences in the viral load. Age was associated with the viral load (r = −0.077; *p* = 0.045), but BMI was not. With regard to symptoms and laboratory values, we did find a few associations with the SARS-CoV-2 viral load. The mean log_10_ viral load was significantly higher in veterans reporting fevers (4.88 vs. 3.96; *p* < 0.0001) and respiratory symptoms (4.67 vs. 3.47; *p* < 0.0001). Individuals with gastrointestinal symptoms had a smaller but statistically significant difference in the viral load (4.71 vs. 4.22; *p* = 0.006). No association was observed between the viral load and the peak CRP, ferritin, D-dimer, or nadir absolute lymphocyte count.

In a univariate comparison, we found that vaccinated veterans had higher log_10_ viral loads (4.71 vs. 4.24; *p* = 0.028) than those who were unvaccinated. However, after adjusting for days from the onset of symptoms to sampling and for the SARS-CoV-2 variant (determined by the predominant variant on the date of sampling, as described above), this association was no longer statistically significant.

### 3.2. Predictors of Moderate-to-Severe COVID-19

We then evaluated the predictive role that the NP SARS-CoV-2 log_10_ viral load had on several different outcomes of moderate-to-severe COVID-19. Given the extensive literature on the other factors that influence the severity of the disease, we examined the association of the following risk factors with our outcomes: age, BMI, diabetes mellitus (DM), coronary artery disease (CAD), chronic obstructive pulmonary disease (COPD)/asthma, immunosuppression, ethnicity, and current tobacco use. The vaccination status was also examined in our analysis, as vaccinations have been shown to directly impact the viral load. Factors with a *p*-value < 0.15 were included in the final model (Table 5). In our final model, we also adjusted for the timing of the sample in relation to the onset of symptoms and also whether or not the sample was obtained during the Delta wave (as we observed higher viral loads during the Delta wave).

#### 3.2.1. Emergency Department (ED) Visit or Hospital Admission

To examine the predictive ability of the viral load on a mild infection requiring an ED visit and moderate-to-severe infections as estimated by infections requiring a hospital admission, we first examined the association of the viral load on this outcome by adjusting for the time of sampling and by the wave of infection (non-Delta vs. Delta). We found that the SARS-CoV-2 viral load was not a significant predictor of ED visits or hospitalization; this model had an AIC of 752.0. We then also included the risk factors for severe diseases that met our pre-specified threshold for inclusion. Only age, BMI, CAD, DM2, and COPD had a *p*-value of < 0.15 and were included in the final model (Table 6). This final model had an AIC of 651.9. The significant predictors in this model were age (*β* = 0.019; *p* = 0.005), BMI (*β* = 0.031; *p* = 0.050), and DM2 (*β* = 1.01; *p* = 0.0002).

#### 3.2.2. Hospital Admission

We then evaluated the role of the viral load on the outcome of hospital admission, adjusting for the day of sampling and the wave of infection. We found that the SARS-CoV-2 viral load again was not a predictor for this outcome. The AIC for this model was 597.2. Using the same process described above, we then included additional risk factors for moderate-to-severe infections that met the criteria in our model. In our final model, we included age, CAD, DM2, COPD/asthma, and BMI along with the viral load and wave of infection. This resulted in an AIC of 514.6 (Table 6). The only significant predictors in this model were age (*β* = 0.038; *p* < 0.0001) and DM2 (*β* = 0.64; *p* < 0.016).

#### 3.2.3. Supplemental Oxygen

We then evaluated the predictive ability of the viral load on the need for supplemental oxygen, again adjusting for the day of sampling and the wave of infection. We again found that the SARS-CoV-2 viral load was not a significant predictor for this outcome, with an AIC of 486.8 for the model. As described above, we then developed a best predictive model that included the additional risk factors of age, CAD, DM2, COPD/asthma, and any immunosuppression, which is shown in Table 6. The final model had an AIC of 455.1, but the only significant predictors in this model were age (*β* = 0.028; *p* = 0.001), day of sampling (*β* = 0.038; *p* < 0.020), and DM2 (*β* = 0.83; *p* < 0.003).

#### 3.2.4. Death during Hospitalization

Finally, we evaluated the predictive ability of the viral load on death during hospitalization, again adjusting for the day of sampling and the wave of infection. Here, the SARS-CoV-2 viral load was significant (*β* = 0.33; *p* < 0.045), with a model AIC of 129.9. As described above, we then developed a best predictive model that included the additional risk factors that met our criteria, which included age, DM2, CAD, and any immunosuppression (Table 6). The final model had an AIC of 112.2 and the only significant predictor in the model was age (*β* = 0.079; *p* < 0.002).

#### 3.2.5. Vaccination Status

Being fully vaccinated was associated with a significantly reduced risk of needing supplemental oxygen (*p* = 0.004) after adjusting for the wave of the epidemic. Full vaccination was not significantly predictive of the other clinical outcomes. When analyzing only unvaccinated people, the predictive factors for the outcomes of hospital admission, supplemental oxygen, and death during hospitalization did not significantly differ from the analysis of the total cohort.

## 4. Discussion

In this cross-sectional study, we examined the association between the SARS-CoV-2 viral load obtained from NP swabs and the clinical outcomes, focusing on moderate-to-severe COVID-19. Numerous other studies have now addressed this question [4,5,8,12,13,17,18,19,20,22], but, in comparison to this study, very few have addressed the timing of the sample in relation to the onset of symptoms or utilized droplet digital PCR, which is substantially more accurate and precise for viral load measurements than standard quantitative real-time PCR. Furthermore, unlike other previous studies, this study included a cross-section of all individuals presenting for testing, ranging from asymptomatic to those requiring admission to the ICU.

As others have previously noted, we found that that the timing of sampling was a key predictor of the viral load, both in relation to the onset of symptoms and in relation to the circulating SARS-CoV-2 variant at the time [41]. The Delta wave was shown to be associated with higher viral loads than prior waves [41,42]. Here, we observed a greater than 1.5 log10 increase during the Delta wave. Numerous reports have also suggested a higher rate of moderate-to-severe disease during the Delta wave than in previous waves. As it was unclear if those differences were due to higher viral loads or other features of the virus, we developed predictive models with and without the wave of infection as a predictor to see if it impacted the importance of the NP viral load as a predictor. We found that the removal of the wave of infection did not significantly impact the importance of the viral load.

A significant proportion of individuals who had a detectable NP viral load remained asymptomatic [43]. This varied by the wave of the SARS-CoV-2 pandemic and also by their vaccination status. After controlling for these two factors and the timing of the sampling, we still found that individuals with a fever and respiratory symptoms had higher NP viral loads than those without those symptoms, suggesting that the early control of replication may be associated with an asymptomatic or a pauci-symptomatic infection. More recent data using oral antivirals have also shown a correlation with the return of the viral load and the return of symptomatology after the initial treatment of the infection [44].

To best evaluate the role of the NP SARS-CoV-2 viral load as a predictor of moderate-to-severe COVID-19, we evaluated four different related outcomes. Notably, in our best predictive models, the viral load only approached a significance as a predictor when the outcome was in-hospital mortality. BMI, CAD, DM2, COPD/asthma, and immunosuppression all showed varying levels of impact on our chosen outcomes, but the strongest and most consistent predictor by far was age. This again corroborated the data from large cohorts [45,46], which also showed age to be the major risk factor for progression to moderate-to-severe COVID-19.

Our cohort was unique in that it included about 25% of individuals who were fully vaccinated. However, we did not see an impact of vaccination on the viral load after adjusting for the wave of infection and for the timing of the sampling from the onset of symptoms. Vaccination clearly prevented a more severe disease and was associated with a significantly reduced risk of in-hospital mortality; thus, it may impact the viral load in the lower respiratory tract differently than the NP.

We recognize several limitations with our study, foremost of which was the number of factors that needed to be controlled for in this cross-sectional study. Although we collected data on most of these factors (e.g., timing from the onset of symptoms, vaccination status, age, and comorbid risk factors), we could only approximate the particular variant from the timing of the sample collection as we did not have sequencing data available for each of the samples. With the large number of factors, we also would have liked to have had a larger number of samples, but, as a single-center study, additional collections would have led to an increase in complexity, given the viral differences in each successive wave of the pandemic.

In conclusion, we found that the NP viral load was predictive of a symptomatic disease and approached a significance for in-hospital mortality. However, with our limited sample size, we were unable to demonstrate that the NP viral load was predictive of moderate-to-severe disease in the pre-Delta and Delta phases of the epidemic. These limited findings still do corroborate prior observations that early viral control may be prevent the progression of disease.

## Figures and Tables

**Figure 1 viruses-15-00446-f001:**
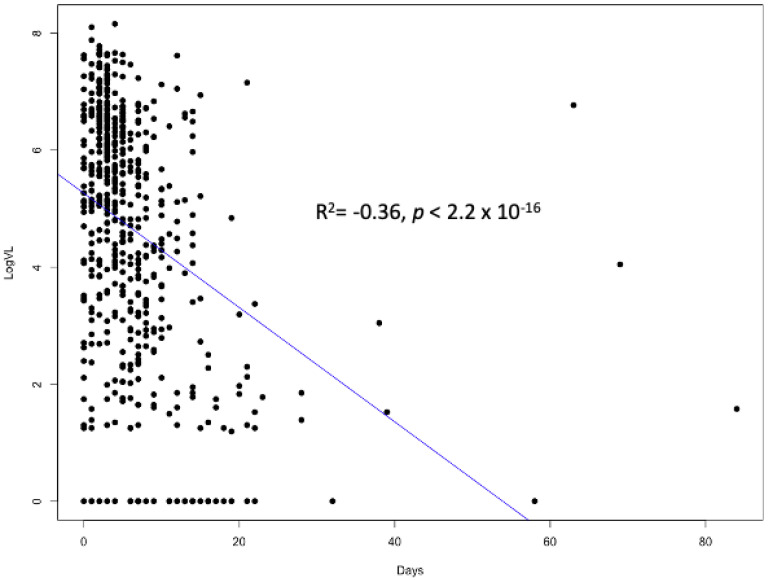
Scatterplot of Log10 SARS-CoV-2 viral load vs. days between onset of symptoms and nasopharyngeal sample collection. Spearman rho = −0.39, with *p*-value < 0.00001.

**Table 1 viruses-15-00446-t001:** Primers and Probe Used in ddPCR.

Label Name	Oligonucleotide Sequence (5′ to 3′)
2019-nCoV_N1-F	GAC CCC AAA ATC AGC GAA AT
2019-nCoV_N1-R	TCT GGT TAC TGC CAG TTG AAT CTG
2019-nCoV_N1-P	FAM-ACC CCG CAT/ZEN/TAC GTT TGG TGG ACC-3IABkFQ

Note: Oligonucleotide sequences were from the Center for Disease Control and Prevention [40].

**Table 2 viruses-15-00446-t002:** Preparation of PCR Mix.

Component	1 × (µL)
Supermix	5
Reverse Transcriptase	2
DTT, 300 mM	1
Forward Primer, 10 µM	1.8
Reverse Primer, 10 µM	1.8
Probe, 2.5 µM	2
Water	5.4
Sample (80–117 ng)	1
Total	20

**Table 3 viruses-15-00446-t003:** Thermal Cycler Conditions.

Temperature, °C	Time	Number of Cycles
50	60 min	1
95	10 min	1
95	30 s (ramp rate 2 °C/s)	40
55	1 min (ramp rate 2 °C/s)	40
98	10 min	1
4	Infinite	1

**Table 4 viruses-15-00446-t004:** Descriptive characteristics of the cohort.

Characteristic	Description	Number with Data Available
Age *	50.2 (+/− 17.1) years	698
Gender		
Male	86.6% (604)	698
Ethnicity		
Hispanic/Latino	23.6% (158)	669
Comorbidity		
DM2	17.4% (121)	697
CAD	7.9% (55)	695
HTN	33.8% (236)	698
Hemodialysis	0.4% (3)	697
Prior solid organ transplant	0.4% (3)	697
COPD/asthma	9.2% (64)	695
BMI *	29.76 (+/− 6.1)	639
Any immunosuppression	4.2% (29)	697
Symptoms		
Respiratory	75.3% (516)	685
Gastrointestinal	46.3% (316)	682
Fever	33.9% (231)	682
Headache	20.9% (146)	698
Loss of taste and/or smell	23.1% (161)	698
Laboratory Values		
Peak CRP **	7.09 (2.58–14.62) mg/dL	153
Peak CPK **	167.0 (67.5–325.5) units/L	79
Peak D-dimer**	0.94 (0.51–1.63) mg/L	155
Peak ferritin **	717 (332–1475) μg/L	135
Absolute lymphocytes (nadir) **	0.90 (0.60–1.43) 10^3^/μL	224

BMI: body mass index; DM2: type 2 diabetes mellitus; CAD: coronary artery disease; COPD: chronic obstructive pulmonary disease; HTN: hypertension; CRP: C-reactive protein; CPK: creatinine phosphokinase. * Mean and standard deviation; ** median and range.

**Table 5 viruses-15-00446-t005:** Univariate logistic regression for COVID-19-related outcomes.

	ED Visit or Admission		Admission		Any Supplemental O_2_ Requirement		Death During Hospitalization	
	*β*	*p*-Value	*β*	*p*-Value	*β*	*p*-Value	*β*	*p*-Value
Age	**0.024**	**<0.0001**	**0.043**	**<0.0001**	**0.041**	**<0.0001**	**0.089**	**<0.0001**
BMI	**0.029**	**0.031**	**0.030**	**0.051**	0.022	0.22	0.023	0.57
DM2	**1.26**	**<0.0001**	**1.27**	**<0.0001**	**1.31**	**<0.0001**	**1.60**	**0.002**
CAD	**0.77**	**0.007**	**1.07**	**0.0003**	**1.15**	**0.0003**	**2.36**	**<0.0001**
COPD/ Asthma	**0.42**	**0.11**	**0.72**	**0.010**	**0.71**	**0.026**	0.85	0.19
Active Smoker	0.030	0.91	−0.29	0.36	0.24	0.46	−0.56	0.59
Immunosuppression	−0.39	0.64	1.00	0.010	**0.67**	**0.14**	**1.21**	**0.12**
Hispanic Ethnicity	0.044	0.81	−0.034	0.87	0.23	0.35	−0.10	0.88

Bold *β* and *p*-values met the *p*-value threshold of <0.15 to be included in the predictive model. ED: emergency department; BMI: body mass index; DM2: type 2 diabetes mellitus; CAD: coronary artery disease; COPD: chronic obstructive pulmonary disease.

**Table 6 viruses-15-00446-t006:** Variables included in the final models, including *p*-values in the multivariate logistic model and final model Akaike information criterion (AIC).

ED Visit or Admission	*p*-Value	Admission	*p*-Value	Any Supplemental O_2_ Requirement	*p*-Value	Death During Hospitalization	*p*-Value
COVID-19 wave	0.26	COVID-19 wave	0.55	COVID-19 wave	0.68	COVID-19 wave	0.11
Days from onset of Sx	0.50	Days from onset of Sx	0.23	**Days from** **onset of Sx**	**0.020**	Days from onset of Sx	0.95
Log_10_ viral load	0.26	Log_10_ viral load	0.21	Log_10_ viral load	0.95	Log_10_ viral load	0.096
**Age**	**0.005**	**Age**	**<0.0001**	**Age**	**0.009**	**Age**	**0.002**
**DM2**	**0.0002**	**DM2**	**0.016**	**DM2**	**0.003**	DM2	0.37
CAD	0.43	CAD	0.60	CAD	0.25	CAD	0.15
COPD/asthma	0.31	COPD/asthma	0.14	COPD/asthma	0.20	Any immunosuppression	0.20
BMI	0.50	BMI	0.12	Any immunosuppression	0.53		
**Final AIC**	**651.9**	**Final AIC**	**514.6**	**Final AIC**	**455.1**	**Final AIC**	**112.2**

ED: emergency department; Sx: symptoms; BMI: body mass index; DM2: type 2 diabetes mellitus; CAD: coronary artery disease; COPD: chronic obstructive pulmonary disease.

## Data Availability

Data supporting the reported results will be shared upon a request to Sanjay Mehta (srmehta@health.ucsd.edu).

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
