# Peer review of "Association between SARS-CoV-2 Viral Load and Patient Symptoms and Clinical Outcomes Using Droplet Digital PCR"

_viruses, 2023, doi:10.3390/v15020446_

Round 1

Reviewer 1 Report

          The authors developed a ddPCR (digital droplet PCR) to measure viral load in patients with COVID-19, and correlated this viral load with severity of disease. They found NP viral load did not appear to be a strong predictor of moderate to severe disease, but was predictive of symptomatic disease and approached significance for in-hospital mortality. Some concerns should be addressed before accepting this manuscript.

Major comments

1.   Since NP viral load only approached but did not reach significance in relation to mortality, the authors should rephrase line 26-27 in the Abstract and in the conclusions.

2.   Another limitation of this study is that the infecting viral variant was not known for each sample. Instead, the authors compare the two period (pre-Delta and during Delta wave), although this was discussed.

3.   Although vaccination did not have an impact on the viral load, it definitely had one on the outcome of the disease. Did the authors analyze the relationship between VP viral load and outcome exclusively in non-vaccinated patients?

4.   It would be interesting to analyze the persistence of high viral load in paired samples as another predictive factor of severity.

Minor comments

5.   Figure 1 is confusing. It says viral load vs days but the axes are inverted.

6.   Line 307: a word is missing: detectable NP viral load.

Reviewer 2 Report

This manuscript analyzed the association of the NP viral loads with the clinical outcomes, which is an interesting question. Because the viral loads depend on the collection timing and the SARS-CoV-2 variants, it is great that the authors have considered these factors when doing the analysis. The limitation of the current study is that they did not have the daily viral loads of each patient. I believe that the changes in viral loads after infection and during treatment would be a more interesting parameter for predicting clinical outcomes of a patient. Because of this limitation, I am quite doubtful on the conclusion that the NP viral loads are predicative of symptomatic disease and of in-hospital mortality.  Maybe a conclusion that the NP viral loads at single timing has no strong association with clinical outcomes is more appropriate.  It is suggested that the authors should reanalyze their data and explain that the observations are not coincident. If the same conclusions are still obtained, the authors should explain more on why that nasopharyngeal viral load did not appear to be a strong predictor of moderate to severe disease, but did appear to be a predictor of in-hospital mortality?

Reviewer 3 Report

In the manuscript "Association Between SARS-CoV-2 Viral Load and Patient 2 Symptoms and Clinical Outcomes Using Droplet Digital PCR", the authors evaluated the association between viral load and clinical outcome.

This reviewer has some questions that are required to be answered and suggestions that may improve the manuscript.

Overall comment: What classifies the period as delta?  It should be explained in the text. I believe that in the introduction there should be more details about when the first sample classified as delta was identified in the country and when it became the "delta wave". Before delta, was there another variant dominating? Other studies that compared the viral load with clinical outcome should be cited in the text. See an example: https://doi.org/10.3390/v14122747.

There should be a viral load comparison before and after Delta wave. Furthermore, little data from the ddPCR results are presented. It should be better explored, as it appears in the title of the manuscript.

Minor comments:

Abstract: According to the journal's rules, the abstract must be a single text, without subheadings. Must be edited.

1. Check the document. There are places with double spaces.

2. Lines 109-123: ddPCR methodology. Thermal cycler conditions and reagent concentration should come as text. Cite the CDC as the source of the primers correctly.

3. Line 146: Conflict of interest must appear at the end of the manuscript

4. When did start the vaccination? Why only 21% of the sampled population was vaccinated? And how many doses?

5. Check nasopharyngeal. Appears as text and as an acronym. Standardize.

Round 2

Reviewer 3 Report

The authors addressed all comments. 

Author Response

Thank you for your comments.